# Biofilm Formation and Expression of Virulence Genes of Microorganisms Grown in Contact with a New Bioactive Glass

**DOI:** 10.3390/pathogens9110927

**Published:** 2020-11-10

**Authors:** Viviane de Cássia Oliveira, Marina Trevelin Souza, Edgar Dutra Zanotto, Evandro Watanabe, Débora Coraça-Huber

**Affiliations:** 1Human Exposome and Infectious Diseases Network—HEID, School of Nursing of Ribeirão Preto, University of São Paulo, Bandeirantes Avenue 3900, Ribeirão Preto 14040-904, São Paulo, Brazil; vivianecassia@usp.br (V.d.C.O.); ewatanabe@forp.usp.br (E.W.); 2Department of Dental Materials and Prostheses, School of Dentistry of Ribeirão Preto, University of São Paulo, Café Avenue S/N, Ribeirão Preto 14040-904, São Paulo, Brazil; 3Vitreous Materials Laboratory—LaMaV, Department of Materials Engineering, Federal University of São Carlos, Rod. Washington Luiz km 235, São Carlos 13565-905, São Paulo, Brazil; marina.trevelin@gmail.com (M.T.S.); dedz@ufscar.br (E.D.Z.); 4Department of Restorative Dentistry, School of Dentistry of Ribeirão Preto, University of São Paulo, Café Avenue S/N, Ribeirão Preto 14040-904, São Paulo, Brazil; 5Department of Orthopedic Surgery, Experimental Orthopedics, Medical University of Innsbruck, Peter‒Mayr-Strasse 4b, 6020 Innsbruck, Austria

**Keywords:** biofilms, gene expression, *Candida albicans*, *Pseudomonas aeruginosa*, *Staphylococcus epidermidis*, bioactive glass

## Abstract

Bioactive glass F18 (BGF18), a glass containing SiO_2_–Na_2_O–K_2_O–MgO–CaO–P_2_O_5_, is highly effective as an osseointegration buster agent when applied as a coating in titanium implants. Biocompatibility tests using this biomaterial exhibited positive results; however, its antimicrobial activity is still under investigation. In this study we evaluated biofilm formation and expression of virulence-factor-related genes in *Candida albicans*, *Staphylococcus epidermidis*, and *Pseudomonas aeruginosa* grown on surfaces of titanium and titanium coated with BGF18. *C. albicans*, *S. epidermidis,* and *P. aeruginosa* biofilms were grown on specimens for 8, 24, and 48 h. After each interval, the pH was measured and the colony-forming units were counted for the biofilm recovery rates. In parallel, quantitative real-time polymerase chain reactions were carried out to verify the expression of virulence-factor-related genes. Our results showed that pH changes of the culture in contact with the bioactive glass were merely observed. Reduction in biofilm formation was not observed at any of the studied time. However, changes in the expression level of genes related to virulence factors were observed after 8 and 48 h of culture in BGF18. BGF18 coating did not have a clear inhibitory effect on biofilm growth but promoted the modulation of virulence factors.

## 1. Introduction

Medical-device-associated infections are mainly caused by microorganisms that are able to self-organize as a biofilm community. The formation of a biofilm is a dynamic process that allows free-living planktonic bacteria to benefit from protection and resistance to drugs and host immune attacks. Biofilm progress involves up- and downregulation of a number of genes, which leads to adhesion, extracellular matrix secretion, and detachment in response to different environmental conditions [1]. Many clinical challenges to treat biofilm-associated infections have been reported, such as manifestation of few symptoms or signs and the presence of resistant microorganisms, which can persist in a slow-growing or even intracellular state [2]. 

Biofilm-associated infection is a common cause of implant failure in dentistry and orthopedics [3]. For instance, some approaches have been proposed in order to control biofilm formation on implants, such as modifying surface topography [4,5,6], surface coating [7], and controlled release of ions and drugs [8]. Lin et al. and Saeed et al. pointed out that materials that carry both antibiofilm and osteogenic capacity are favorable for orthopedic applications; however, this research field is relatively new, and suitable materials are not widely available [9,10].

Besides the traditional use of bioactive glass to improve osteointegration [11] new research domains involving this material have emerged [12]. Bioactive glass F18 (BGF18 (SiO_2_–Na_2_O–K_2_O–MgO–CaO–P_2_O_5_)) is a promising newly developed glass with high bioactivity and a wider range of workability when compared to other bioactive glass compositions [13]. Biological tests, involving BGF18, demonstrated tissue proliferation and regeneration with improved biointeraction [14]. Due to its improved workability, coating metallic implants with BGF18 was a feasible process and a promising feature for clinical applications. In in vivo studies, BGF18 coatings have improved the wettability and enhanced bone–implant contact and bone density after two weeks post-implantation [15]. In a direct contact, BGF18 presented a bactericidal property against planktonic microorganisms [16]; nonetheless, studies involving its action on biofilm formation and changes on the morphological and ultrastructural characteristics on the pattern of gene expression were not yet performed. Investigation of virulence genes involved in cell adhesion, biofilm formation, and quorum sensing could clarify whether surfaces coated with BGF18 might interfere with biofilm development. 

Some genes have been suggested to be determinant for biofilm formation. In *Candida albicans* biofilms, the agglutinin-like sequence (*ALS*) that encodes large cell-surface glycoproteins is related to the adhesion mechanism [17]. Hyphal wall protein (*HWP*) is a hyphal-specific adhesion gene that encodes the hyphal cell wall protein promoting adhesion to different surfaces [18]. Hydrolytic aspartyl proteinase (*SAP*) contributes to both adhesion and invasion processes through the degradation or distortion of cell surface structures [19]. In *Staphylococcus epidermidis* biofilms, *icaADBC* (intercellular adhesion and extracellular matrix) has been correlated to primary bacterial adhesion and intercellular adhesion, both necessary for biofilm formation [20]. In *Pseudomonas aeruginosa* biofilms, polysaccharide synthesis (*psl*) is required for adhesion and maintenance of the biofilm structure after the first attachment [21]. Acyl-homoserine-lactone synthase (*lasI*) is connected to quorum sensing signal molecule production [22]. The elastase structural gene (*lasB*) causes tissue damage and degradation of components of the innate immune system [23]. Secreted pyocyanin (*phz*) is related to pyocyanin production, a virulence biomarker of *P. aeruginosa* [24].

For the aforementioned reasons, we consider it essential to clarify whether BGF18 coatings exhibit antibiofilm activity and could modulate biofilm-forming gene expression. Thus, the objective of this study is to evaluate in vitro the biofilm formation on titanium coated with BGF18 and assess its capability to modulate the expression of virulence-factor-related genes in *C. albicans*, *S. epidermidis,* and *P. aeruginosa* biofilms. The null hypothesis of this study is that biofilm growth and gene expression will be similar on titanium and BGF18-coated titanium.

## 2. Results

### 2.1. Antimicrobial Activity

Concerning the biofilm recovery on titanium and BGF18 specimens, there were no statistical differences on the antibiofilm activity of the coatings after 8, 24, and 48 h of culture. All groups had similar biofilm growth in both surfaces in all time points (Table 1, Table 2 and Table 3).

### 2.2. Biofilm Morphology

The coating characteristic of BGF18 is shown in Figure 1. It is possible to observe the deposition of the biomaterial as droplets on the titanium surface (black arrow). In addition, coverage does not appear to prevent bacterial colonization, since it was possible to observe directly the colonization on the BGF18 aggregates (yellow arrows).

### 2.3. pH Variation

The pH values ranged from 6 to 8 in all samples during all the time points tested (Figure 2a–c). The wells with BGF18 samples showed pH values slightly higher in comparison with control samples at early biofilm growth. For *S. epidermidis*, the pH value remained higher during all time points tested; however, this variation was not sufficient to impact biofilm formation. Since the pH values were similar to those that normally prevail in the physiological conditions, statistical comparisons were not conducted. 

### 2.4. Expression of Virulence Genes

The relative quantification method was used to compare the quantitative polymerase chain reaction (qPCR) results. This method was considered acceptable since the efficiencies of both target and reference genes were between 90% and 105%. The mean of the cycle quantification (Cq) for each target gene was normalized to the mean of the Cq of the reference gene amplified from the corresponding sample (ΔCq). ΔΔCq values for BGF18 were calculated based on the difference of the titanium gene expression levels at each time point. Finally, potential transcript level alteration on BGF18 after 8, 24, and 48 h of culture were expressed by 2^−ΔΔCq^, estimating how many times it was higher or lower than the titanium group (expression in control group = 1) (Figure 2d–f). 

*ALS1* and *HWP1* genes, which are related to adhesion mechanism in *C. albicans*, had their expression increased in BGF18 after 8 h of culture (*p* = 0.021). Similar expression increase was observed in *pslA* gene, after 8 h of culture, in *P. aeruginosa* (*p* = 0.021). All virulence-factor-related genes in *C. albicans* (*ALS1*, *HWP1*, and *SAP5*), *S. epidermidis* (*icaADBC*), and *P. aeruginosa* (*lasl*, *pslA*, *lasB*, and *phzH*) were higher expressed in BGF18 after 48 h of culture (*p* = 0.021). 

## 3. Discussion

In this study, the antibiofilm activity of BGF18 applied as a coating on titanium surfaces was evaluated. The assessment also took into consideration the effect on the expression of biofilm virulence genes of the different strains. Based on the results, the null hypothesis was rejected, since there was statistical difference between the BGF18 and non-covered titanium. 

All currently used metallic implants do not bond chemically to bone. Therefore, suitable surface roughness is useful for mechanical interlocking and improvements in biocompatibility and bioactivity [25]. Bioglass coated implants have been largely studied due to its osteogenic properties related to hydroxyapatite layer formation [11]. In this study, the bioactive glass coating features were chosen due to previous in vivo results that have shown improved and speedier osseointegration, as presented by Soares and collaborators [15]. 

In addition to the osteogenic and antibacterial properties, bioglasses have also been investigated for their potential antibiofilm activity [26,27]. In our study however, the biomaterial did not show satisfactory antibiofilm activity (Figure 1c–h). The same result trend was also reported by Xie et al., who examined the rate of infection with *Staphylococcus aureus*, and Begum et al., who evaluated the antibacterial efficacy on *Escherichia coli* and *S. aureus* [28,29]. These authors stated that bioglass 45S5 did not present a significant statistical difference regarding bacterial growth inhibition in in vivo and in vitro tests, respectively. Xie et al. indicated that the antibacterial effect in vitro is concentration dependent, and in their study, particles under 63 µm were used in concentrations from 2.5 to 10 mg/mL [28]. Here however, we did not use pure bioglass particles but a coated surface by the deposition of a small amount of BGF18, about 25% of the total area, as shown in Figure 1a. This would represent a concentration of approximately 1 mg/mL (bioglass/medium). Coraça-Huber et al. found that bioactive glass S53P4 can suppress *S. aureus* biofilm grown on titanium discs in vitro by using a concentration of 500 mg/mL in contact with the titanium discs (particles < 45 µm) [26]. This amount is massively higher than the one used in this study. Although the material configuration was chosen due to its good results regarding osteostimulation and osseointegration [15], the bioactive glass concentration was likely too low to promote antibiofilm action. Coating implants with bioglass normally have the main function of promoting a faster osseointegration, and the coating thickness has a direct effect on stress concentration in the cortical bone for dental implants [30].

No significant variation in pH of the environment containing the BGF18 presence was observed (Figure 2a–c). We consider that the biomaterial concentration used in this study may have been too low to alter the pH of the medium. The increase in pH is associated with the release of alkali and alkali earth ions (i.e., Na and Ca) from the glass into the fluid, which are replaced by H^+^ or H_3_O^+^ ions in the glass structure [31]. Begum et al. showed that the pH levels are directly related to the concentrations of bioglass particles [29]. Literature has linked the antimicrobial activity of bioglasses to an initial pH increase, and the absence of antibiofilm activity of BGF18 in this study could be associated with the small variation in alkalinity [26,31]. In higher concentrations, BGF18 presented an increase in pH either in phosphate-buffered saline (PBS) or simulated body fluid (SBF) solutions and mineralized a hydroxyapatite layer after only 4 to 12 h, as shown by Souza et al. and Gabbai-Armelin et al. [13,14]. It is important to highlight that hydroxyapatite is frequently used as a biocompatible material with no antimicrobial activity [32], and its formation on the BGF18 surface during the early stages of biofilm formation could justify the non-identified statistical difference between titanium and titanium covered with BGF18 groups. Furthermore, the distinct initial pH among different strains might be explained by metabolic changes underlying biofilm formation since microorganisms can trigger different adaptive metabolic pathways according glucose and carbohydrates metabolism [33]. Arce Miranda et al. reported that at a slightly acidic pH, the biofilm formation by *S. aureus* was 3.5-fold higher than at the basic pH [34]. Likewise, we suggest that *S. epidermidis* needs a lightly acid pH for optimal biofilm development.

Both cell adhesion and biofilm formation are controlled by a number of factors that include environmental aspects, bacterial species, surface composition, and essential gene products [35]. It has been suggested that roughness, surface free energy, and hydrophobicity have a significant influence on cell attachment and biofilm formation. Microbial colonization appears to increase as the surface roughness increases. This is because shear forces decrease, and the surface area increases on rougher surfaces [36]. The samples used in this study were coated with BGF18 particles of approximately 50 µm, which could be responsible for a substantial roughness and wettability alteration of the surface, as shown by Soares et al. [15]. Moreover, the precipitation of hydroxyapatite would increase sample roughness, which could potentially affect microorganism’s adhesion and biofilm formation.

Souza et al. reported a broad-spectrum antibacterial property of BGF18, virtually eliminating 100% of planktonic *S. aureus*, *S. epidermidis*, *P. aeruginosa*, and *E. coli* in 24 h of direct contact to the material [16]. The apparent discrepancy between our results and those observed by Souza et al. could be related to the experimental protocol and the exposure pattern to the biomaterial [16]. The authors added the bacteria in direct contact with BGF18 at a concentration of 50 mg/mL. We believe that a higher pH alteration would be noticed in this circumstance, as a result of the biomaterial concentration. Here, the biomaterial was immobilized in a titanium surface in a concentration 50 times lower. Facing this scenario, the pH changes would be less expressive, which could explain the divergence in the results and the absence of antibiofilm activity. It is important to reiterate that the antimicrobial susceptibility of microorganisms in biofilm-associated and non-biofilm-associated states is vastly distinct. Biofilm’s tolerance to antimicrobial agents is about 100–1000 times greater compared to that of the planktonic form [37]. According to Cabal et al. the interface between glass and bacterial cell membrane is crucial for biocidal effect [38]. As the BGF18 coating covered 25% of total area of the specimen, the contact between the biomaterial and cells membranes was compromised. Herein, the insufficient amount of material for the coating layer can justify the absence of antibiofilm activity, since a 60% covering of F18 glass particles on the steel surface was responsible for bactericidal and antibiofilm activity against *S. aureus* and methicillin-resistant *S. aureus* (MRSA) [39]. In our study, probably, the bacterial cell adhesion had started on the free surface, and after the formation of the mineralized hydroxyapatite layer, it reached the remaining areas.

Based on inconsistency between the results of this study and the already announced antibacterial property of BGF18, we decided to investigate whether virulence-factor-related genes could be differently expressed in biofilms grown on BGF18 surface. We hypothesized whether BGF18 could act selectively, in the early stages of the biofilm growth, favoring bacterial cells with increased biofilm formation capacity. This question was addressed taking into consideration that the capacity of bacterial adhesion and biofilm phenotype can be influenced by the chemical properties of the adhesion surface [40]. Although the culture conditions used were not aimed to increase the virulence of the microorganisms, the conditions tested may have jeopardized the antibacterial capacity of the biomaterial. 

An elevated expression of adhesion-mechanism-related genes was observed in BGF18 after 8 h of culture, in *C. albicans* (*ALS1* and *HWP1*) and *P. aeruginosa* (*pslA*) biofilms. It could be suggested that BGF18 offered an initial limitation to cell adhesion, that led microorganisms to activate adherence-related genes. In *S. epidermidis*, we propose that the bacteria did not demand activation of the virulence-factor-related genes because BGF18 did not offer a limitation to biofilm formation. Thomas and Wigneshweraraj demonstrated that bacteria can coordinate the expression of their virulence determinants according to adaptive response in replaying environmental alterations, such as concentrations of ions and pH [41]. The virulence factors, which play an important role in the bacterial colonization, survival, and capacity of tissue invasion, depend on a large number of cell-associated and extracellular factors [42]. After 48 h of culture all virulence-factor-related genes in *C. albicans*, *S. epidermidis*, and *P. aeruginosa* were higher expressed in BGF18. This result suggest that microorganisms activated the genes in response to the ambient characteristics. The time-dependent surface modification of BGF18 might specifically affect virulence-factor-related genes. 

Similar to the results observed in this study, Chang et al. observed an upregulation of virulence factors, as a defense mechanism in *P. aeruginosa*, in responses to oxidative stress [43]. Sarkisova et al. demonstrated that calcium addition resulted in thicker *P. aeruginosa* biofilms with increased alginate and extracellular protease secretion [42]. The literature has proposed that high phosphate concentration can alter virulence factors [44]. Here, the calcium and phosphate ions released by BFG18 to the medium, and its post consumption for hydroxyapatite layer formation, as demonstrated by Souza et al., could have reached a concentration high enough to promote modulation of the virulence factors [15]. 

Bacterial adhesion and biofilm formation are remarkable issues in medical-device-associated infections. Lin et al. and Saeed et al. reinforced that the ideal implant surface modification should have a long duration of anti-infective effects, mechanical stability, and host biocompatibility [9,10]. Furthermore, the studies that evaluated the antimicrobial action of surface modified material had been conducted only in vitro. It is necessary to translate then into to animal models in order to justify clinical recommendations in the future. To date, clinical impacts of biofilms in orthopedic infections have been increasingly recognized and the modulation of virulence factors should be carefully analyzed. Further in vivo models involving biofilm-associated infections could give us new insights to understand the microorganism and host response in the presence of BGF18. 

In vitro tests for biofilm growth on biomaterial surfaces are frequently carried out using culture medium, considered a rich cell growth supplement. It does not exactly reproduce the in vivo conditions where the host’s tissue environment is susceptible to several changes in constitution and immune responses. The methodology used in this study may alter biological responses and not simulate the reality of tissue surroundings. As the BGF18 coating used in this study is directed to orthopedical applications, further studies could evaluate the alteration of morphological and ultrastructural characteristics, such as the pattern of protein expression beyond biofilm formation under the same composition of human synovial fluid and bone cells metabolites, for example. In this case, the nutrient supply during the pathogenesis of periprosthetic joint infections and the interaction of the bacterial cell with host synovial fluid should also be considered. 

This study is the first step toward enhancing our understanding of biofilm growth in contact with a BGF18-coated surface. Taken together, our findings suggest that different coating strategies should be investigated in order to maintain the antibacterial activity observed previously in planktonic bacterial growth. As a limitation of the study, maybe the CFU counts and the time points evaluated were not sensitive enough to assess alterations in biofilm growth according the subtle pH variation in the used concentration of the biomaterial. In addition, the analysis using such a small concentration (1 mg/mL), may not have achieved the minimum inhibitory concentration for the glass to be effective and statistically significant. 

## 4. Materials and Methods 

### 4.1. Sample Preparation

The fabrication and manufacturing processes of BGF18 were described by Souza et al. [45]. For the coating process, the same methodology described by Soares et al. was used [15]. Briefly, the biomaterial powder was deposited on the surface of 54 circular titanium samples (10 × 4 mm) by the pneumatic atomization technique. A coating layer (25% of area) was applied on both sides of the specimens. The particle size had a mean diameter of 50 μm. Non-coated titanium samples were used as control. Both samples surfaces were standardized in 400# sandpaper. For sterilization, the specimens were inserted in a hot air oven (Memmert, Schwabach, Germany) at 160 °C for 2 h. 

### 4.2. Culture Conditions

The antibacterial activity of the BGF18-coated samples was tested against three different strains from the American Type Culture Collection (ATCC): *C. albicans* (10231), *S. epidermidis* (12228), and *P. aeruginosa* (27853). The bacteria inoculums (10^6^ colony-forming units (CFU)/mL) were prepared from exponentially growing cultures in Tryptic Soy Broth (BD Difco Sparks, MN, USA). The specimens were randomly assigned in 24-well tissue culture plates (Greiner Bio-One, Kremsmünster, Austria), and 2 mL of medium broth containing standardized cell suspension was added to each well. Hence, the ratio of bioglass mass and the medium used in this study was of approximately 1 mg/mL. The plates were incubated at 37 °C on a shaker at 120 cycles/min (Edmund Bühler GmbH, Bodelshausen, Germany). The experiments were carried out in duplicate taking into consideration two different time points.

### 4.3. Antimicrobial Activity 

Antimicrobial activity against biofilm formation was evaluated after 8, 24, and 48 h of incubation. After incubation, the specimens were taken from the wells, washed in fresh PBS to remove the planktonic cells and transferred to a 15 mL tube (VWR International, Radnor, Pennsylvania, USA) containing 2 mL of new PBS. The tubes were sonicated (Bandelin GmbH, Berlin, German) for 3 min at 100% intensity for the disruption of the biofilms. Serial dilutions (10^−1^ to 10^−6^) of the suspension were seeded in agar plates containing different culture media according to each microorganism (*C. albicans*—Sabouraud Dextrose Agar (BD Difco); *S. epidermidis* and *P. aeruginosa*—Müller Hinton Agar (BD Difco)). The plates were incubated at 37 °C for 24 h and the number of CFU/mL was determined, based on a dilution providing 1‒300 colonies. The values of count of CFU/mL were converted to log_10_.

### 4.4. Biofilm Morphology

To evaluate both specimens’ characteristics and biofilm morphology, samples were evaluated by scanning electron microscopy. After 48 h of incubation, biofilms samples were fixed with 2.5% glutaraldehyde for 60 min and subsequently dehydrated in a graded ethanol series (30%, 50%, 70%, 90%, and 100%). Samples without biofilm growth did not undergo fixation and dehydration steps. Specimens were then examined under high vacuum with a JEOL JSM-35CF microscope (Tokyo, Japan). 

### 4.5. pH Measurement

Readings of pH were taken using paper strips (Sigma Aldrich, Saint Louis, Missouri, USA) after each incubation time point (8, 24, and 48 h). Briefly, 50 µL of the supernatant of both control and BGF18 culture wells was dropped on paper strips, and the color change was registered. 

### 4.6. Activity on the Expression of Virulence Genes

The expression of virulence-factor-related genes in *C. albicans*, *S. epidermidis*, and *P. aeruginosa* biofilms were assessed after 8, 24, and 48 h of culture. The full description of the evaluated genes and their functions are presented in Table 4.

After each incubation time point, the discs were removed from the wells, washed in fresh PBS, and transferred to a new plate. One milliliter of Trizol (Sigma Aldrich) was deposited directly on the discs. After successive pipetting for the disruption of the biofilms from the surfaces, the suspension was transferred to a microtube and homogenized by means of a tissue tearer (Biospec Products Inc., Bartlesville, EUA) during 20 s. The homogenization process was repeated three times, and the samples were placed on ice between repetitions. After homogenization, 300 µL of chloroform (Sigma Aldrich) was added, and the mixture was centrifuged at 12,000× *g* for 5 min at 4 °C. The supernatant was collected, and the RNA was precipitated with isopropyl alcohol (Sigma Aldrich) and centrifuged again as described above. The RNA was washed twice with 500 µL 75% ethanol (Sigma Aldrich) and air dried. For quantification and qualification of the isolated RNA, the samples were resuspended in 20 µL of ultrapure DEPC-treated water and measured by a spectrophotometer (Thermo Scientific, Waltham, Massachusetts, USA), with readings at 230, 260, and 280 nm. The integrity of the RNA was attested by agarose gel electrophoresis (Agargen, Madrid, Spain).

For cDNA synthesis, 1 µg of RNA template and a high-capacity cDNA reverse transcription kit (Thermo Scientific) were used. The reaction was performed in a thermal cycler (Ampliterm, Madison, Wisconsin, EUA), following the manufacturer’s instructions. For quantification of the gene expression, 10 µL of IQ SYBR Green Supermix (Bio-Rad, Hercules, California, USA), 6 µL of distilled water, 1 µL of primer forward (Sigma Aldrich), 1 µL of primer reverse (Sigma Aldrich), and 1 µL of cDNA template, standardized at 50 ng/µL were used. The reaction was carried out by using a PikoReal real-time PCR system (Thermo Scientific) as follows: one cycle at 95 °C for 3 min, 40 cycles at 95 °C for 15 s, and at 60 °C for 1 min, with a final melting curve analysis from 65 to 95 °C with increments of 0.5 °C every 5 s. The real-time PCR amplifications were conducted in triplicate, and the threshold values were set as automatic. Slopes of standard curves (10-fold dilution series) were applied to indicate PCR efficiency. All primers and genomic sequences used in this study are listed in Table 5. 

### 4.7. Statistical Analysis

Data were tested to respectively adhere to the normal and homogeneous distribution by Shapiro–Wilk and Levene tests. Differences in the biofilm recovery on both titanium and BGF18 surfaces, according to the different testing times, were analyzed by the T Student and Mann–Whitney U tests. The relative expression values in BGF18 were compared according to the expression level in titanium, at the same time point, by the Mann–Whitney U test. All tests were carried out through the IBM SPSS Statistics 25.0 software (IBM Corp Armonk, NY, USA). The significance level was set at 0.05.

## 5. Conclusions

Although promising results were previously noticed involving the antibacterial property of BGF18 in the planktonic stage, our observations demonstrated that the biomaterial did not have a clear inhibitory effect on *C. albicans*, *S. epidermidis*, and *P. aeruginosa* biofilm growth. The relevance of the present study highlights the need for extensive investigation regarding the antimicrobial and antibiofilm characteristics of BGF18 surface coating. Under the experimental conditions tested and comparing with other author´s results, we demonstrated that the amount of biomaterial seem be crucial for a broad antibiofilm activity. Even though further studies are needed to evaluate biological responses when simulating a tissue reality, the BGF18 coating changed the gene expression after simulation of a young and mature biofilm, suggesting that the biomaterial coating promoted a modulation of virulence factors.

## Figures and Tables

**Figure 1 pathogens-09-00927-f001:**
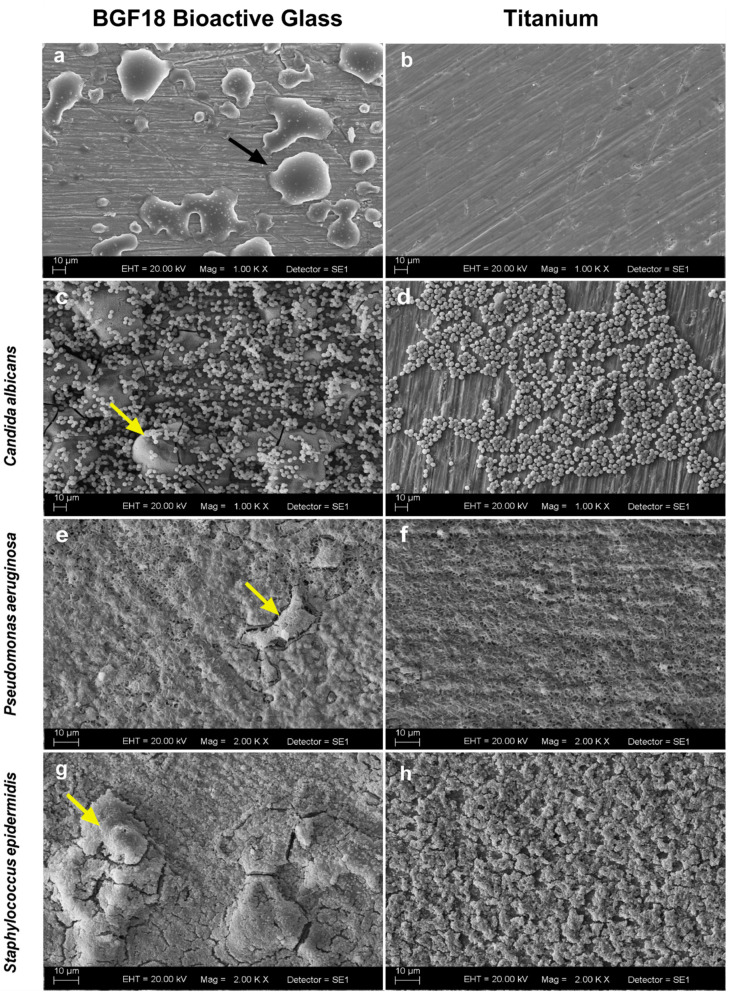
(**a**,**b**) Scanning electron microscopy of a titanium specimen covered with BGF18 (magnification 1000×) and titanium (magnification 1000×), respectively. (**c**–**h**) Biofilms at 48 hours of *C. albicans* ((**c**,**d**), magnification 1000×), *P. aeruginosa* ((**e**,**f**), magnification 2000×), and *S. epidermidis* ((**g**,**h**), magnification 2000×), growth on titanium specimen covered with BGF18 and titanium surfaces. Black arrow indicates BGF18 deposition on the titanium surface, while yellow arrows indicate the directly colonization on the BGF18 aggregates.

**Figure 2 pathogens-09-00927-f002:**
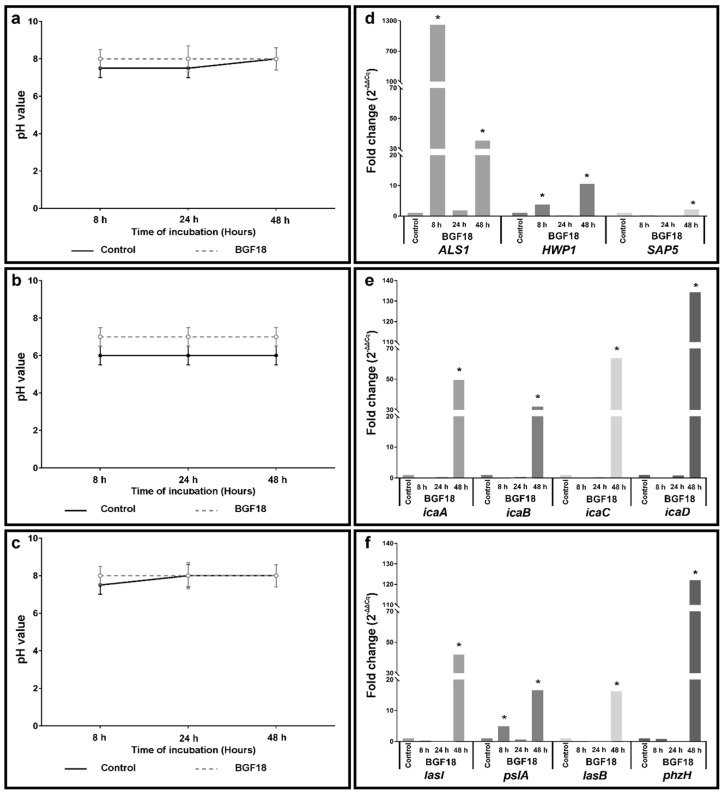
(**a**–**c**) pH variation of *C. albicans* (**a**), *S. epidermidis* (**b**), and *P. aeruginosa* (**c**) biofilms after cultivation (8, 24, and 48 h) in contact with titanium (control) and titanium covered with BGF18. (**d**–**f**) Relative gene expression (fold change—2^−∆∆Cq^) in *C. albicans* (**d**), *S. epidermidis* (**e**), and *P. aeruginosa* (**f**) biofilms after cultivation (8, 24, and 48 h) in contact with titanium covered with BGF18. The relative expression values in titanium covered with BGF18 were compared according to the expression level in titanium (control) at the same time point. * Difference was statistically significant (*p* = 0.021) in BGF18 after 8 h (*C. albicans*—*ALS1*, *HWP1*; *P. aeruginosa*—*pslA*) and 48 h of culture (*C. albicans*—*ALS1*, *HWP1*, *SAP5*; *S. epidermidis*—*icaADBC*; *P. aeruginosa*—*lasl*, *pslA*, *lasB*, *phzH*). Comparisons between titanium (control) and titanium covered with BGF18 were carried out through Mann–Whitney U test.

**Table 1 pathogens-09-00927-t001:** Colony-forming units (log_10_^CFU/mL^) of *Candida albicans* after 8, 24, and 48 h of incubation on the surface of titanium and titanium covered with bioactive glass F18 (BGF18).

Incubation Time	Group				*p*-Value
	Titanium		Titanium Covered with BGF18		
	Mean ± Standard Deviation (Median)	95% Confidence Interval (Minimum–Maximum)	Mean ± Standard Deviation (Median)	95% Confidence Interval (Minimum–Maximum)	
8 h	5.87 ± 0.15 (5.83)	5.64; 6.10 (5.74–6.08)	5.78 ± 0.08 (5.78)	5.65; 5.92 (5.68–5.88)	0.330 *
24 h	6.03 ± 0.42 (6.17)	5.36; 6.69 (5.41–6.35)	5.96 ± 0.19 (5.99)	5.65; 6.26 (5.70–6.14)	0.771 *
48 h	5.60 ± 0.39 (5.69)	4.98; 6.21 (5.09–5.91)	4.92 ± 0.99 (5.39)	3.35; 6.50 (3.44–5.47)	0.200 ^†^

* T Student test. ^†^ Mann–Whitney U test.

**Table 2 pathogens-09-00927-t002:** Colony-forming units (log_10_^CFU/mL^) of *Staphylococcus epidermidis* after 8, 24, and 48 h of incubation on the surface of titanium and titanium covered with BGF18.

Incubation Time	Group				*p*-Value
	Titanium		Titanium Covered with BGF18		
	Mean ± Standard Deviation (Median)	95% Confidence Interval (Minimum–Maximum)	Mean ± Standard Deviation (Median)	95% Confidence Interval (Minimum–Maximum)	
8 h	6.89 ± 0.16 (6.89)	6.64; 7.14 (6.71–7.06)	7.05 ± 0.15 (7.06)	6.81; 7.29 (6.86–7.20)	0.190 *
24 h	6.56 ± 0.07 (6.57)	6.45; 6.67 (6.47–6.64)	6.60 ± 0.31 (6.58)	6.12; 7.09 (6.30–6.96)	0.790 *
48 h	7.03 ± 0.09 (7.02)	6.88; 7.18 (6.95–7.14)	7.20 ± 0.19 (7.15)	6.90; 7.50 (7.05–7.45)	0.157 *

* T Student test.

**Table 3 pathogens-09-00927-t003:** Colony-forming units (log_10_^CFU/mL^) of *Pseudomonas aeruginosa* after 8, 24, and 48 h of incubation on the surface of titanium and titanium covered with BGF18.

Incubation Time	Group				*p*-Value
	Titanium		Titanium Covered with BGF18		
	Mean ± Standard Deviation (Median)	95% Confidence Interval (Minimum–Maximum)	Mean ± Standard Deviation (Median)	95% Confidence Interval (Minimum–Maximum)	
8 h	7.36 ± 0.14 (7.32)	7.13; 7.59 (7.24–7.57)	7.39 ± 0.20 (7.41)	7.08; 7.70 (7.15–7.60)	0.823 *
24 h	7.02 ± 0.06 (7.00)	6.91; 7.12 (6.95–7.11)	7.11 ± 0.14 (7.15)	6.88; 7.33 (6.90–7.23)	0.279 *
48 h	6.71 ± 0.44 (6.66)	6.01; 7.41 (6.26–7.26)	6.72 ± 0.11 (6.73)	6.55; 6.89 (6.59–6.84)	0.963 *

* T Student test.

**Table 4 pathogens-09-00927-t004:** Genes evaluated in *C. albicans*, *S. epidermidis*, and *P. aeruginosa* biofilms.

Microorganism	Gene Name	Gene Description	Function	Reference
*C. albicans*	*ALS1*	Agglutinin-like sequence 1	Encodes cell-surface glycoproteins that are involved in adhesion of fungal cells to host and abiotic surfaces.	[17]
	*HWP1*	Hyphal wall protein 1	Associated with adhesive functions necessary for biofilm integrity, attachment to host, and virulence.	[18]
	*SAP5*	Secreted aspartyl proteinase 5	Degrades host cell proteins, contributing to tissue damage and invasion.	[19]
*S. epidermidis*	*icaADBC* operon	Intercellular adhesion	Mediates intercellular adhesion of bacterial cells by synthesis of polysaccharide intercellular adhesin (PIA).	[20]
*P. aeruginosa*	*lasl*	Acyl-homoserine-lactone synthase	Related to quorum sensing mechanism. Required for the synthesis of N-(3-oxododecanoyl)homoserine lactone.	[22]
	*pslA*	Polysaccharide synthesis	Involved in attachment to surfaces and extracellular polysaccharide biosynthesis.	[21]
	*lasB*	Elastase structural B	Secretion of extracellular proteases that cleaves host proteins, favoring pathogenesis of infection.	[23]
	*phzH*	Pyocyanin biosynthesis	Acts converting the phenazine-1-carboxylic acid into phenazine in the pyocyanin synthetic pathway.	[24]

**Table 5 pathogens-09-00927-t005:** Primer sequence used.

Microorganism	Gene	Sequence 5′–3′	Amplification Size (pb)
*C. albicans*	*ALS1*	TTCTCATGAATCAGCATCCACAA (*F*) CAGAATTTTCACCCATACTTGGTTTC (*R*)	53
	*HWP1*	GCTCAACTTATTGCTATCGCTTATTACA (*F*) GACCGTCTACCTGTGGGACAGT (*R*)	67
	*SAP5*	CAGAATTTCCCGTCGATGAGA (*F*) CATTGTGCAAAGTAACTGCAACAG (*R*)	78
	*ACT1* (Reference)	GCTGGTAGAGACTTGACCAACCA (*F*) GACAATTTCTCTTTCAGCACTAGTAGTGA (*R*)	87
*S. epidermidis*	icaA	CTCTTGCAGGAGCAATCAAT (*F*) AGAGCACGTGGTTCGTACTT (*R*)	176
	*icaD*	GAGGCAATATCCAAGGTAA (*F*) AAATTTCCGTGTTTTCAACATT (*R)*	194
	*icaB*	AATGGCTTAAAGCACACGAC (*F*) AAACAGGAAAGGCATTGTCA (*R*)	137
	*icaC*	TATAGGCGTCGGAATGATGT (*F*) TCCAGTTAGGCTGGTATTGG (*R*)	100
	*16s rDNA* (Reference)	GAAAGCCACGGCTAACTACG (*F*) CATTTCACCGCTACACATGG (*R*)	203
*P. aeruginosa*	*lasI*	GGCTGGGACGTTAGTGTCAT (*F*) AAAACCTGGGCTTCAGGAGT (*R*)	104
	*pslA*	TCCCTACCTCAGCAGCAAGCTGGT (*F*) CGGATGTCGTGGTTGCGTACCAGGTAT (*R*)	198
	*lasB*	AGACCGAGAATGACAAAGTGGAA (*F*) GGTAGGAGACGTTGTAGACCAGTTG (*R*)	81
	*phzH*	TGCGCGAGTTCAGCCACCTG (*F*) TCCGGGACATAGTCGGCGCA (*R*)	214
	*rpsL* (reference)	GCAACTATCAACCAGCTGGTG (*F*) GCTGTGCTCTTGCAGGTTGTG (*R*)	231

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
