# Peer review of "Biofilm Formation and Expression of Virulence Genes of Microorganisms Grown in Contact with a New Bioactive Glass"

_pathogens, 2020, doi:10.3390/pathogens9110927_

Round 1

Reviewer 1 Report

line 66, 71 & 78: C. albicans --> Candida albicans, S. epidermidis --> Staphylococcus epidermidis, P. aeruginosa --> Pseudomonas aeruginosa

lines 82-83: Candida albicans, Staphylococcus epidermidis and Pseudomonas aeruginosa --> C. albicans, S. epidermidis and P. aeruginosa

line 81: in vitro --> in vitro

Table 1: The legend indicates that the table reports both mean and median, but only one of these is presented.

Figure 2, legend: B-H --> C-H

line 118: C. albicans --> C. albicans

Figure 3: T1 is not defined. This figure need to be larger. The text in the figure is not readable without a magnifying glass.

lines 140: in vivo --> in vivo

lines 142-158: What microorganisms were tested by Xie et al., Begum et al.and Coraca-Huber et al.? Passos et al. (Bactericidal activity and biofilm inhibition of F18 bioactive glass against Staphylococcus aureus Materials Sciencd & Engineering C) would fit in this discussion.

lines 178-179: "Also, the precipitation of the hydroxyapatite layer increased the coated sample roughness ..." It must be made clear that this is not referring to the present study. A suggestion: Precipitation of hydroxyapatite would increase sample roughness which could affect cell adhesion and biofilm formation.

line 202: virulenceof --> virulence of

line 235: In vitro --> In vitro

lines 297-298: newly centrifuged: centrifuged again as described above ?

Author Response

line 66, 71 & 78: C. albicans --> Candida albicans, S. epidermidis --> Staphylococcus epidermidis, P. aeruginosa --> Pseudomonas aeruginosa

 lines 82-83: Candida albicans, Staphylococcus epidermidis and Pseudomonas aeruginosa --> C. albicans, S. epidermidis and P. aeruginosa

 line 81: in vitro --> in vitro

 line 118: C. albicans --> C. albicans

lines 140: in vivo --> in vivo

 line 202: virulenceof --> virulence of

 line 235: In vitro --> In vitro

Response: The presentation of microorganism´s names and latin word were given in italic form. As well attention was directed to genus abbreviation.

 Table 1: The legend indicates that the table reports both mean and median, but only one of these is presented.

Response: Tables were redesigned in order to show all central tendency measures.

Figure 2, legend: B-H --> C-H

 Figure 3: T1 is not defined. This figure need to be larger. The text in the figure is not readable without a magnifying glass.

 Response: Figures were reconsidered, and a new proposal was presented in the reviewed manuscript.

lines 142-158: What microorganisms were tested by Xie et al., Begum et al.and Coraca-Huber et al.? Passos et al. (Bactericidal activity and biofilm inhibition of F18 bioactive glass against Staphylococcus aureus Materials Science & Engineering C) would fit in this discussion.

Response: We appreciate the comment and add the requested information.

Text changes:

The same result trend was also reported by Xie et al. that examined the rate of infection with Staphylococcus aureus and Begum et al. who evaluated the antibacterial efficacy on Escherichia coli and S. aureus.”

"Coraça-Huber et al. found that the bioactive glass S53P4 can suppress S. aureus biofilm grown on titanium discs in vitro by using a concentration of 500 mg/mL in contact with the titanium discs (particles < 45 µm) [26]."

Herein, the insufficient amount of material for the coating layer can justify the absence of antibiofilm activity, since F18 glass particles covering 60% of the steel surface was responsible for bactericidal and antibiofilm activity against S. aureus and methicillin-resistant S. aureus (MRSA) (Passos et al., 2021).”

 lines 178-179: "Also, the precipitation of the hydroxyapatite layer increased the coated sample roughness ..." It must be made clear that this is not referring to the present study. A suggestion: Precipitation of hydroxyapatite would increase sample roughness which could affect cell adhesion and biofilm formation.

 lines 297-298: newly centrifuged: centrifuged again as described above?

Response: We appreciate the comment and rewrote the sentences.

Reviewer 2 Report

The research article by Oleveira et al entitle “Biofilm formation and expression of virulence genes of microorganisms grown in contact with a new bioactive glass” put forth new results from the bioglass coating to titania and its influence on biofilm related virulence associated genetic changes. Paper is technically sound and scientifically valid, i.e. the methods are appropriate and properly conducted. Paper needs major revision with respect to clarify the gaps in method adopted for literature search,  citing some latest report on the section discussed herein, organization of figures, results drawn and conclusion made. Authors are strongly suggested to include proper literature background and references suggested.  My comments are appended below:

On page 2, paragraph 2 line 66-78, the information about biofilm related genes can be presented in a tabular form for the conveniences of the readers to quickly comprehend the role of each genes in biofilm related virulence.

Author missed some seminal papers on the topics discussed herein this report. For example, Along with sentence “For instance, some approaches have been proposed in order to control biofilm formation on implants, such as modifying surface topography”, please cite along with reference 4 report  https://doi.org/10.1371/journal.pone.0175428 & DOI https://doi.org/10.1088/1758-5082/4/2/025001 which was first study with TiO2 and antibacterial copper whiskers associated topography modification to inhibit bacterial colonization and biofilm destabilization.

The sentence before the last one is extremely long and written in confusing way, please rephrase and split the sentence to understand it.

Table a as presented is unclear what are different values mentioned in brackets, please present data in 3 different tables for 8, 24 and 48 hrs to clearly understand. Why Mann Whitney U test was performed on 48 hours samples only and

The SEM micrograph in figure 2.a-b are at different magnification, so it makes difficult to exactly compare the surface topography of the 2 substrate (titanium vs glass), can author provide images with same magnification to comprehend  it better?

In figure 2, what are the drop like observation in the deposition of the biomaterial as droplets on the titanium surface? Is that local aggregation of Titanium due to electron beam from SEM? Why aggregates evolved and how it can influence the biofilm characteristics.

In Figure 3., the genes mentioned inside the box cannot be read, please plot it with different color combination to comprehend it better or explain the data in figure legends. In addition, can authors represent with start mark or any other sign the statistically different expression mentioned in plots of figure 3?

Page 6, line 159 onwards, can authors e more precise about explaining the influence of pH variation on biofilm formation in relation with bioglass explaining what are protonating/deprotonating (H+) functional groups in bioglass which might influence the pH changes?

Minor

Many typos and inappropriate punctuations exist, manuscript needs a thorough check of the English language and need fix these minor but important punctuation errors. Some examples are cited below:

Typos:

anti-biofilm proprieties

in table 1 legend, add gap in bacterial designation: C.albicans, S.epidermidis

inconsistent sps representation, sometime italicized and sometimes not, please cross check whole draft. For example in figure 1, C. albicans, S. epidermidis and P. aeruginosa. Same with the in vitro and in vivo.

Increasement shall be increase or increment

hydrogenionic

There are many abbreviations which are unclear and had never explained in full form through author provided a brief list at the end. Authors are suggested provide a list of abbreviation used before introduction to make draft suitable to read.

Author Response

On page 2, paragraph 2 line 66-78, the information about biofilm related genes can be presented in a tabular form for the conveniences of the readers to quickly comprehend the role of each genes in biofilm related virulence.

Response: We appreciate the suggestion and added a table in Materials and Methods section.

Text change: “The expression of virulence factors related-genes in C. albicans, S. epidermidis and P. aeruginosa biofilms were assessed after 8, 24 and 48 h of culture. The full description of evaluated genes and their functions are presented in table 4.”

Table 4. Genes evaluated in C. albicans, S. epidermidis and P. aeruginosa biofilms.

Microrganism

Gene name

Gene description

Function

Reference

C. albicans

ALS1

Agglutinin-like sequence 1

Encodes cell-surface glycoproteins that are involved in adhesion of fungal cells to host and abiotic surfaces.

[17]

HWP1

Hyphal wall protein 1

Associated with adhesive functions necessary for biofilm integrity, attachment to host and virulence.

[18]

SAP5

Secreted aspartyl proteinase 5

Degrades host cell proteins, contributing to tissue damage and invasion.

[19]

S. epidermidis

icaADBC operon

Intercellular adhesion

Mediates intercellular adhesion of bacterial cells by synthesis of polysaccharide intercellular adhesin (PIA).

[20]

P. aeruginosa

lasl

Acyl-homoserine-lactone synthase

Related to quorum sensing mechanism. Required for the synthesis of N-(3-oxododecanoyl)homoserine lactone.

[22]

pslA

Polysaccharide synthesis

Involved in attachment to surfaces and extracellular polysaccharide biosynthesis.

[21]

lasB

Elastase structural B

Secretion of extracellular proteases that cleaves host proteins, favoring pathogenesis of infection.

[23]

phzH

Pyocyanin biosynthesis

Acts converting the phenazine-1-carboxylic acid into phenazine in the pyocyanin synthetic pathway

[24]

Author missed some seminal papers on the topics discussed herein this report. For example, Along with sentence “For instance, some approaches have been proposed in order to control biofilm formation on implants, such as modifying surface topography”, please cite along with reference 4 report  https://doi.org/10.1371/journal.pone.0175428 & DOI https://doi.org/10.1088/1758-5082/4/2/025001 which was first study with TiO2 and antibacterial copper whiskers associated topography modification to inhibit bacterial colonization and biofilm destabilization.

The sentence before the last one is extremely long and written in confusing way, please rephrase and split the sentence to understand it.

Response: As suggested the references were added do the reviewed manuscript and the sentence was rephrased.

Table a as presented is unclear what are different values mentioned in brackets, please present data in 3 different tables for 8, 24 and 48 hrs to clearly understand. Why Mann Whitney U test was performed on 48 hours samples only and

Response: Data regarding biofilm growth rates were presented isolated for each microorganism. Moreover, we added all central tendency measures in order to clarify the understanding. The Mann Whitney U test was performed on Candida albicans after 48 hours of biofilm growth because this group did not adhere to normal and homogeneous distribution verified by Shapiro Wilk and Levene tests.

The SEM micrograph in figure 2.a-b are at different magnification, so it makes difficult to exactly compare the surface topography of the 2 substrate (titanium vs glass), can author provide images with same magnification to comprehend  it better? In figure 2, what are the drop like observation in the deposition of the biomaterial as droplets on the titanium surface? Is that local aggregation of Titanium due to electron beam from SEM? Why aggregates evolved and how it can influence the biofilm characteristics.

Response: We appreciate the comment and reanalyzed all the figures. A new proposal was presented in the reviewed manuscript. We apologize about our initial mistake, which identification of Titanium and Bioactive Glass surfaces were exchanged.

In Figure 3., the genes mentioned inside the box cannot be read, please plot it with different color combination to comprehend it better or explain the data in figure legends. In addition, can authors represent with start mark or any other sign the statistically different expression mentioned in plots of figure 3?

Response: A new proposal was presented in the reviewed manuscript.

Page 6, line 159 onwards, can authors be more precise about explaining the influence of pH variation on biofilm formation in relation with bioglass explaining what are protonating/deprotonating (H+) functional groups in bioglass which might influence the pH changes?

Response: The explanation was added to reviewed manuscript.

Text change: “The increase in pH is associated with release of alkali and alkali earth ions (i.e. Na and Ca) from the glass into the fluid, which are replaced by H+ or H3O+ ions in the glass structure [31].”

Minor

Many typos and inappropriate punctuations exist, manuscript needs a thorough check of the English language and need fix these minor but important punctuation errors. Some examples are cited below:

Typos:

anti-biofilm proprieties

in table 1 legend, add gap in bacterial designation: C.albicans, S.epidermidis

inconsistent sps representation, sometime italicized and sometimes not, please cross check whole draft. For example in figure 1, C. albicans, S. epidermidis and P. aeruginosa. Same with the in vitro and in vivo.

Increasement shall be increase or increment

hydrogenionic

There are many abbreviations which are unclear and had never explained in full form through author provided a brief list at the end. Authors are suggested provide a list of abbreviation used before introduction to make draft suitable to read.

Response: The presentation of microorganism´s names and latin word were given in italic form. As well attention was directed to explain the abbreviations the first time it appears in the main text.

Reviewer 3 Report

The submitted manuscript focuses on important issue of antimicrobial properties of bioactive glasses. However, there is some questions, which should be discussed.

1) Fig. 1 bring any important information. Especially, that there is no discussion about it and there is no error bars or another indicators if the differences are statistically important. Please extend this section and explain the differences in initial pH among different strains. 

2) Please indicate the novelty of the study. Please refer to previous research carried out by the authors, e.g. [14], show the originality and purposefulness of research in the submitted article.

3) Fig. 2: the description in text suggest that yellow arrows shown aggregates  on the BGF18, but they are in column with Titanium samples. Where is the mistake? 

4) Dle conclusions should be more extended. 

5) Please provide all latin names in italic.

Author Response

1) Fig. 1 bring any important information. Especially, that there is no discussion about it and there is no error bars or another indicators if the differences are statistically important. Please extend this section and explain the differences in initial pH among different strains. 

Response: A new proposal for figure 1 was presented in the reviewed manuscript. We added error bar; however, since the pH values were similar to those that normally prevails in the physiological conditions any statistical comparisons were conducted. In addition, a potential explanation about the differences in initial pH among different strains were added in discussion section.

Text changes:

Results: “The pH values ranged from 6 to 8 in all samples during all the time points tested (Figure 3a - c). The wells with BGF18 samples showed pH values slightly higher in comparison with control samples at early biofilm growth. For S. epidermidis the pH value remained higher during all time points tested; however, this variation was not sufficient to impact biofilm formation. Since the pH values were similar to those that normally prevails in the physiological conditions any statistical comparisons were conducted.”

Discussion: “Furthermore, the distinct initial pH among different strains might be explained by metabolic changes underlying biofilm formation since microorganism can trigger different adaptive metabolic pathways according glucose and carbohydrates metabolism [33]. Arce Miranda et al. reported that at a slightly acidic pH, the biofilm formation by S. aureus was 3.5-fold higher than at the basic pH [34]. Likewise, we suggest that S. epidermidis needs a lightly acid pH for an optimal biofilm developed.”

2) Please indicate the novelty of the study. Please refer to previous research carried out by the authors, e.g. [14], show the originality and purposefulness of research in the submitted article.

Response: The novelty of the study was highlighted in introduction and discussion sections.

Text changes:

Introduction: “BGF18 presented a bactericidal property against planktonic microorganisms [16]; nonetheless, studies involving its action on biofilm formation and changes on the morphological and ultrastructural characteristics on the pattern of gene expression were not yet performed. Investigation of virulence genes involved in cell adhesion, biofilm formation and quorum sensing could clarify whether surfaces coated with BGF18 might interfere on biofilm development.

Discussion: “This study is the first step towards enhancing our understanding of biofilm growth in contact with BGF18-coated surface. Taken together, our findings suggest that different coating strategies should be investigated in order to maintain the antibacterial activity observed previously in planktonic bacterial growth.”

3) Fig. 2: the description in text suggest that yellow arrows shown aggregates  on the BGF18, but they are in column with Titanium samples. Where is the mistake? 

Response: We appreciate the comment and reanalyzed all the figures. A new proposal was presented in the reviewed manuscript. We apologize about our initial mistake, which identification of Titanium and Bioactive Glass surfaces were exchanged.

4) Dle conclusions should be more extended. 

Response: As suggested we rewrote the conclusion.

Text change: “Although promising previously results was noticed involving antibacterial property of BGF18 in planktonic stage, our observations demonstrated that biomaterial did not have a clear inhibitory effect on C. albicans, S. epidermidis and P. aeruginosa biofilm growth. The relevance of the present study highlights the need for extensive investigation regarding antimicrobial and antibiofilm characteristics of BGF18 surface coating. Under the experimental conditions tested and comparing with other author´s results, we demonstrated that the amount of biomaterial seem be crucial for a broad antibiofilm activity. Even though further studies are needed to evaluate biological responses when simulating a tissue reality, BGF18 coating changed the gene expression after simulation of a young and mature biofilm, suggesting that the biomaterial coating promoted a modulation of virulence factors.”

5) Please provide all latin names in italic.

Response: The presentation of microorganism´s names and latin word were given in italic form.

Round 2

Reviewer 2 Report

Accept

Reviewer 3 Report

Thank you for your kind response.